# Identification and Expression Analysis of *Dsx* and Its Positive Transcriptional Regulation of *IAG* in Black Tiger Shrimp (*Penaeus monodon*)

**DOI:** 10.3390/ijms232012701

**Published:** 2022-10-21

**Authors:** Wen-Ya Wei, Jian-Hua Huang, Fa-Lin Zhou, Qi-Bin Yang, Yun-Dong Li, Song Jiang, Shi-Gui Jiang, Li-Shi Yang

**Affiliations:** 1Key Laboratory of South China Sea Fishery Resources Exploitation & Utilization, Ministry of Agriculture and Rural Affairs, South China Sea Fisheries Research Institute, Chinese Academy of Fishery Sciences, Guangzhou 510300, China; 2Shenzhen Base of South China Sea Fisheries Research Institute, Chinese Academy of Fishery Sciences, Shenzhen 518108, China; 3College of Fisheries and Life Science, Shanghai Ocean University, Shanghai 201306, China; 4Key Laboratory of Efficient Utilization and Processing of Marine Fishery Resources of Hainan Province, Sanya Tropical Fisheries Research Institute, Sanya 572018, China

**Keywords:** *Penaeus monodon*, *Dsx*, insulin-like androgenic gland hormone, transcriptional regulation

## Abstract

*Doublesex* (*Dsx*) is a polymorphic transcription factor of the *DMRTs* family, which is involved in male sex trait development and controls sexual dimorphism at different developmental stages in arthropods. However, the transcriptional regulation of the *Dsx* gene is largely unknown in decapods. In this study, we reported the cDNA sequence of *PmDsx* in *Penaeus monodon*, which encodes a 257 amino acid polypeptide. It shared many similarities with Dsx homologs and has a close relationship in the phylogeny of different species. We demonstrated that the expression of the male sex differentiation gene *Dsx* was predominantly expressed in the *P. monodon* testis, and that *PmDsx* dsRNA injection significantly decreased the expression of the insulin-like androgenic gland hormone (*IAG*) and male sex-determining gene while increasing the expression of the female sex-determining gene. We also identified a 5′-flanking region of *PmIAG* that had two potential cis-regulatory elements (CREs) for the *PmDsx* transcription. Further, the dual-luciferase reporter analysis and truncated mutagenesis revealed that *PmDsx* overexpression significantly promoted the transcriptional activity of the *PmIAG* promoter via a specific CRE. These results suggest that *PmDsx* is engaged in male reproductive development and positively regulates the transcription of the *PmIAG* by specifically binding upstream of the promoter of the *PmIAG*. It provides a theoretical basis for exploring the sexual regulation pathway and evolutionary dynamics of Dmrt family genes in *P**. monodon*.

## 1. Introduction

Insulin-like androgenic gland hormone (*IAG*), secreted by the androgenic glands (AG) [1], is hypothesized to play a role in male sexual differentiation and maintenance of sexual characteristics [2]. Since *IAG* was first identified in a decapod (the red-clawed crayfish *Cherax quadricarinatus*) in 2007, it has also been found in many crustacean species such as *Macrobrachium rosenbergii* [3]; *Macrobrachium nipponensis* [4,5]; *Penaeus monodon* [6]; *Fenneropenaeus chinensis* [7], and *Portunus pelagicus* [8]. The mechanism by which *IAG* mediates male sexual differentiation and gonad development has been demonstrated in crustaceans [9,10]. For example, silencing of *IAG* expression with siRNA [11] and dsRNA [12] in *M. rosenbergii* causes functional sex reversal and stagnation of testicular spermatogenesis. Meanwhile, *EsIAG* knockdown may produce partial “neo-female” in the males of the *Eriocheir sinensis* [13]. In addition, it has been shown that *IAG* is engaged in the initiation and maintenance of key reproductive hormones that regulate the formation of sexually dimorphic traits. For instance, in *Scylla paramamosain*, crustacean female sex hormone (*CFSH*) positively regulates female germinal pore formation and inhibits the expression of *Sp-IAG* in males by controlling *STAT* (Signal Transducer and Activator of Transcription), suggesting that *IAG* is involved in the transcriptional mechanisms of sex regulation [14].

*Doublesex* (*Dsx*) is a polymorphic transcription factor that was first identified in *Drosophila melanogaster* and shown to control sexual dimorphism at different developmental stages [15]. In different organisms, sex-specific *Dsx* has distinct and conserved effects on sexual characteristics, such as yolk protein (*Yp*) transcription [16], male courtship songs [17], male-specific sexual combs, and female-specific fertilization sacs [18]. In insects such as *D. melanogaster*, the *Dsx* gene has sex-specific splicing in males and females, generating male (Dsx M) and female-specific (Dsx F) Dsx isoforms [19]. Dsx F and Dsx M have opposing effects and compete with each other to regulate specific genes [20]. *Dsx* plays an integral role in the sexual differentiation and development of different insect species. In previous studies, a transgenic TALENs system was used to target the female-specific isoform (Bmdsx F) and the male-specific isoform (Bmdsx M) of the *Bmdsx* gene in *Bombyx mori*, resulting in female/male-specific sterility [18,21].

There is no splicing of *Dsx* genes and no splice variant forms in nematodes, echinoderms, vertebrates, and arachnids. Interestingly, in the crustacean *Daphnia magna*, two types of *Dsx* (*Dsx1* and *Dsx2*) were found to be involved in male trait development. Knockdown of *DapmaDsx1* in male embryos results in the production of female traits including ovarian maturation, while ectopic expression of *DapmaDsx1* in female embryos leads to the development of a male-like phenotype. However, silencing and overexpression of *Dsx2* did not cause any significant phenotypic changes [22]. Moreover, as a crustacean, only one *Dsx* gene has been identified in the *M. rosenbergii* (*MrDsx*) and has been implicated in its potential role in the regulation of molt, growth, and sexual differentiation. Furthermore, crustaceans *Dsx*/*Dmrt* was reported to play the role in the regulation of the *IAG* gene expression by binding the promoter region to affect development and maintain male sexual characteristics [23,24,25].

The black tiger shrimp (*Penaeus monodon*) (Crustacea: Decapoda: Prawns), is an economically significant crustacean species that is widely farmed along the southern coast of China. *P. monodon* has significant sexual dimorphism whereby females outgrow males. Though there have been many efforts to reveal the process of sexual dimorphism [26,27,28], the expression and function of critical genes that participate in sex determination and differentiation remain in deep research. In this study, we identified and characterized sex-determining homologs of the *Dsx* gene (*PmDsx*) from *P. monodon* and investigated its temporal and spatial sexual dimorphism expression profiles. Furthermore, we report that *PmDsx* binds directly to the cis-regulatory element (CRE) within the *PmIAG* promoter in vitro; it confirmed its positive regulatory role in activating the male-biased gene to participate in the process of sex determination and differentiation of *P. monodon* in vivo.

## 2. Results

### 2.1. cDNA Cloning and Sequence Analysis of PmDsx

The *PmDsx* mRNA sequence is 1458 bp in length (GenBank: MT521744.1) and the Open Reading Frame (ORF) is from positions 518 to 1291 bp (774 bp). This nucleotide sequence translated to a peptide sequence of 257 amino acids with a calculated molecular weight of 28.214 kDa. No signal peptide and glycosylation site but four phosphorylation sites were detected in this amino acid sequence, including Tyrosine (Y56), Threonine (T147), Serine (S22), and Serine (S50). ExPASy tool revealed that PmDsx protein possessed a conserved DM domain between amino acids 148 and 196 (Figure 1). The three-dimensional model obtained showed that the DM domain was composed of several zinc finger structures and contained an α-helix and two anti-parallel β-sheets. There was a pair of cysteine residues at the N-terminal and a pair of histidine residues at the C-terminal of the zinc fingers, which had formed a cavity to accommodate exactly one Zn^2+^ (Figure 2).

### 2.2. Multiple Sequence Alignment and Phylogenetic Analysis of PmDsx

Basic Local Alignment Search Tool (BLAST) analysis revealed that the complete amino acid sequence of *PmDsx* shared high sequence identity with *Dsx* sequences from other Decapoda; the results showed that the predicted protein sequence had the highest homology with *F. chinensis* (94.55% similarity), followed by *Penaeus japonicus* (87.11% similarity), *M. rosenbergii* (67.50% similarity), *C. quadricarinatus* (45.97% similarity), and *Sagmariasus verreauxi* (37.87% similarity), which all contain a highly conserved DM structural domain (Figure 3A). To assess the evolutionary relationship of PmDsx with Dsx of other species, phylogenetic trees of Dsx protein sequences of several species were generated using the NJ method. The results showed that the evolutionary tree of Dsx was divided into two main clusters: crustaceans and insects. Among the crustaceans, *P. monodon* clustered into a small branch with *F. chinensis* and *P. japonicus*, and belongs to a large branch with *C. quadricarinatus*, *S. verreauxi*, *M. rosenbergii*, and *Portunus trituberculatus*. The insects branch included *C**. quinquefasciatus*, *Anopheles Stephens*, and *Megaselia scalaris* (Figure 3B).

### 2.3. Tissue Distribution and Sexually Dimorphic Expression of PmDsx

Based on EF-1ɑ as a control, qRT-PCR was used to detect the spatial expression pattern of *PmDsx* in 12 tissues (Figure 4). The results showed that *PmDsx* was expressed in all tissues, with the highest expression level in testis, followed by the epidermis, muscle, gill, and lymph, low expression in the heart, hepatopancreas, stomach, nerve, and lowest expression in spermatophore, intestine, and ovary. (*p* < 0.05). Moreover, we examined the expression of *PmDsx* in individuals at different developmental stages. We used the whole shrimp with 2 cm body length because their gonads did not grow yet, but we can distinguish their sex by qPCR method of our lab, and then used the testis of male shrimp and ovary of female shrimp with 13 or 26 cm body length for qPCR analysis [26]. The results showed that the expression of *PmDsx* in the reproductive system of juvenile shrimp (2cm), young shrimp (13~15cm), and mature shrimp (21~26cm) showed significant sex dimorphism. The expression in male individuals and reproductive systems was significantly higher than that in female individuals and reproductive systems and decreased with the maturation of the testis. Similarly, the expression of *Dsx* was higher in female larvae and decreased significantly with the development of the ovary (Figure 5).

### 2.4. RNA Interference of PmDsx Reduces the Transcript Level of Sex-Related Genes

The interference efficiency and target gene expression were detected by qRT-PCR. The results showed that *PmDsx* expression was significantly decreased at 48 h, indicating that RNA interference was effective. In addition, we explored the effect of *PmDsx* knockout on the following sex-related genes: *IAG*, *SRY*-related HMG-box gene 9 (*Sox9*), and Forkhead transcription factor L2 (*Foxl2*). The results are shown in the figure when *PmDsx* was silenced, the expression of *PmDsx* in dsDsx group was significantly down-regulated compared with PBS group and dsGFP group at 2, 4, and 6 days, and the dsDsx group at 6 days showed a rebound in *PmDsx* expression compared to the dsDsx group at 2, 4 days. Moreover, the expression trends of *PmIAG* genes were basically the same as of *PmDsx*. The expression of *PmSox9* expression in the dsDsx group was lower than in the dsGFP group at 4 days and lower than in the PBS group at 4, and 6 days. In 2, and 4 days, the expression of *PmSox9* was slightly decreased. On the contrary, the expression of *PmFoxl2* was significantly increased compared with PBS group and dsGFP group at 2, 4, and 6 days, also *PmDsx* expression was remarkably upregulated in the dsDsx group at 2, 4, and 6 days compared to 0 days (Figure 6).

### 2.5. Prediction of Transcription Factor Binding Sites in the Promoter Region of PmIAG

The promoter containing a 2000 bp sequence upstream of the translation start site of the *IAG* in the *P. monodon* genomic sequence (LOC119596132) was obtained in NCBI. Potential transcription factor binding sites were predicted by online software, and the analysis showed that the *IAG* promoter region contains a series of transcription factors binding sites such as *Sox2*/*B2*, *Pax5*, *Myf3*, *C/EBPbeta*, and *Dmrt1*, and importantly, there are two predicted *Dsx* binding located at −623 to −633 bp and −692 to −702 bp regions (Figure 7). Based on the location of the binding sites, we then successfully constructed several truncated mutants that contained different regions of promoter sequences (Figure 8 and Appendix A). Four promoter fragments were amplified from *P. monodon* genomic DNA by PCR using specific primers (Table 1). All cloned *PmIAG* 5′-flanking sequences were validated and used for dual-luciferase reporter analysis.

### 2.6. PmDsx Regulates the Transcriptional Activity of PmIAG

Four truncated mutants were cotransfected into HEK293T cells with the *PmDsx* constructs to identify the functional transcription factor binding site of *PmDsx* within the promoter of *PmIAG* and using the empty vector as a control. Luciferase activity assays confirmed that *PmDsx* overexpression significantly enhanced the expression activity of the promoter region from −374 to −938 bp compared to the control. However, no significant changes were observed for *PmDsx* overexpression in other regions. These data suggested that *PmDsx* is indeed bound to a promoter region between −374 bp to −938 bp that contains two *Dsx* binding sites to regulate the translation activities of the *PmIAG* gene in *P. monodon* (Figure 9).

## 3. Discussion

*Dsx* is widely conserved among arthropods as well as vertebrates and was regarded as the master-switch gene in the sex determination cascade in different organisms. *PmDsx* contained a classic DM structural domain of the *Dsx* family, which is highly conserved in the *Penaeus*. The three-dimensional structure confirmed *PmDsx* had a DNA binding domain (DM domain) in N-terminal and a dimerization domain (*Dsx* dimer) in C-terminal. The DM domain as a zinc module utilizes its conserved amino acid residues for zinc chelating and promoter binding [27], while the *Dsx* dimer as an alpha-helical motif improves DNA binding and encourages *Dsx* dimerization [28]. The DM domain may have specialized and enhanced functions in mediating relevant physiological activities, making the specific role played by *Dsx* in reproduction.

*PmDsx* had a high identity and close phylogenetic relationship with *F. chinensis*, as both contain only one Dsx spliceosome but are far from other branches of arthropods such as *Drosophila* and insects that have two spliceosomes. The male-specific Dsx M and female-specific Dsx F was directed splicing by tra and the initiation-lethal Sxl protein, and regulate the downstream genes for sex-specific differentiation [29,30]. Since Dsx is highly conserved in the Drosophila sex determination cascade pathway and can determine individual sex, we hypothesize that the *Dsx* gene in *P. monodon* and other decapods is involved in the sex determination and differentiation process, but its function is partially lost or altered during the long arthropod genetic differentiation process. This is the reason for the generation of different numbers of Dsx spliceosomes.

The expression of *PmDsx* has an obvious sex dimorphism that is significantly high in the testis and lowest in the ovary not only in the tissue but also in the different development stages, suggesting its potentially important role in the development and maintenance of the male reproductive system. *F**.chinensis* and *Charybdis feriatus* also exhibit similar sex differences in that their transcripts are mainly expressed at the highest levels in male-specific structures during embryonic development [25,31]. Interestingly, in *C. quadricarinatus*, *Cqdsx* was strongly expressed in the female gonads and a strong hybridization signal was observed mainly in oocytes and ovarian lamellae, and a weaker hybridization signal was detected in spermatocytes suggesting that *Cqdsx* plays an important role in female ovarian development and differentiation [32]. The *Spdsx* expression in tissues of the *Scylla paramamosain* was similar to that of *C. quadricarinatus*, but different from that of *C. feriatus*, indicating that the *Dsx* expression pattern is highly variable among species [33]. Simultaneously, the differential expression of *Dsx* in individuals of different genders is significant, and in general, the temporal and spatial expression of a gene is tightly linked to its effects. Our study showed that the total expression of *PmDsx* was always higher in male individuals than in female individuals during different developmental periods, illustrating a sex-specific role of *PmDsx* in the male development of *P. monodon*. The regulation of *Dsx* on downstream somatic sex differentiation of terminal genes is thought to be related to transcriptional regulation, and ultimately leads to the establishment of sexual dimorphism in adult Drosophila [34]; therefore, the study of the regulatory mechanism of *Dsx* on downstream target structural genes seems to be essential.

Subsequently, we successfully silenced the expression of *PmDsx* by RNA interference technique. Different from our previous studies [35], we explored not only the positive regulation of *PmDsx* on *PmSox9* and *PmIAG* but also the negative regulation of *PmFoxl2*. Some studies have shown that disruption of XY fish *Dmrt1* leads to increased expression of *Foxl2* and *cyp19a1a* in Nile tilapia. In contrast, *Foxl2* deficiency in XX fish displayed varying degrees of oocyte degeneration or even complete sex reversal, and elevated expression of *Dmrt1* and *Cyp11b2* illustrated the antagonistic role of *Dmrt1* and *Foxl2* in estrogen secretion-induced sex differentiation in tilapia [36]. Concurrently, *Dmrt1* can positively regulate the transcription of *sox9b* by binding directly to a specific CRE within the *Sox9b* promoter in the testis [37], which seems to be consistent with our study. Hence, we can hypothesize that *Dmrt*, *Foxl2,* and *Sox9* may have a regulatory function in sex determination and differentiation, at least in teleost fishes. Moreover, as with shrimp, *MniDMRT11E* RNAi leads to significant positive regulation of *IAG* transcription in the male abdominal ganglion of *M. nipponense* [38]. Coincidentally, the *MroDMRT11E* gene of *M. rosenbergii* has the same positive regulatory effect on *IAG* [39], which indicates that *DMRT* might be available to regulate the expression of *IAG* in shrimp. Considering the important participation of *IAG* in male crustaceans and the confirmation of several studies [40,41], we hypothesized that PmDsx might be directly or indirectly involved in the upstream pathway of PmIAG.

As a key transcription factor, *IAG* plays a pivotal role in sex redistribution [12,42], and spermatogenesis [2,41] in reproductive crustaceans. In a study of the relevance of osmoregulatory genes to reproductive development in *E. sinensis*, the expression of *IAG* is determined by the phosphorylation of transcription factors such as *iDMY* and *Sox15* by PKA in the gonad, and these phosphorylated transcription factors bind upstream of the promoter of *IAG*, thereby inducing its transcription [43]. These data suggest that *IAG* may be a conserved factor downstream of the sexual differentiation process in decapods, which can be regulated by upstream *DMRT* and *Sox* family genes. In this study, analysis of the *PmIAG* promoter sequence predicts two putative *Dsx*/*mab-3* binding sites at −623 bp and −692 bp in the *PmIAG* gene sequence, a *DMRT1* binding site in front of its promoter region start codon, and three *Sox* family gene binding sites (*Sox2*, *SoxB2*, *Sox11*). In 1999, Yi reported DNA sequences for the *Mab-3* and *Dsx* binding sites, with AATGTTGCGAT (A)NT for the Mab-3 binding site and nTnGT (A)ACAATGTT (A)nCC for the *Dsx* binding site [44], which overlap with our predicted AATGTAG for the *Dsx* transcription factor binding site, validating the accuracy of this predicted binding site. As an upstream transcription factor, *Dsx* could bind to the promoter region of *IAG* to regulate its transcription in *M. rosenbergii* [23] and *F. chinensis* [25]. Therefore, we can tentatively speculate that *PmDsx* may link to a binding site in the upstream region of the *IAG* promoter to trigger its transcription.

Several truncated mutants, which contained different promoter regions with different lengths based on the location of transcription factor binding sites, confirmed the regulatory effect of *PmDsx* on *PmIAG* expression by a dual-luciferase reporter assay, revealing that *PmDsx* was successfully bound to *PmIAG* and triggered its transcriptional activity. Similarly, *F. chinensis FcDsx* could bind at −434 bp to −416 bp (including the binding site) of the *FcIAG* gene and regulate its expression [25]; our joint results hypothesize that *Dsx* and its target gene, *IAG*, play an integral role in the shrimp sex-regulatory cascade pathway.

## 4. Materials and Methods

### 4.1. Experimental Animals and Cell Culture

All healthy cultured *P. monodon* were collected from the experimental base of the South China Sea Fisheries Research Institute in Shenzhen (Guangdong, China). The shrimps were maintained in recirculating seawater tanks with a salinity of 29‰, a temperature of 24–28 °C, and a pH of 7.5–7.8 before use. The shrimp with a weight of 25.46 ± 2.5 g and a length of 12.63 ± 0.72 cm were used for tissue expression and RNA interference in this study. Moreover, the human embryonic kidney-derived HEK293 cell line was cultured in Dulbecco’s Modified Eagle’s medium (DMEM) (Gibco original, origin Australia), supplemented with 10% heat-inactivated fetal bovine serum (FBS) (Invitrogen, Waltham, MA, USA), 100 U mL−1 penicillin, and 100 µg mL−1 streptomycin at 37 °C in a humidified atmosphere containing 5% CO_2_, as described previously (Lemaitre et al., 1996). Logarithmic growth phase cells were seeded into 24-well plates and cultured for 24 h, then grown until they reached 70–80% confluency.

The larval development period of *P. monodon* was determined according to the method of Huang [26]. Starting from the fertilized eggs, the larval development period was confirmed through observation under the microscope. Samples were taken once in each period until the larvae. Thirty sex-determined *P. monodon* seedlings were selected and observed under an anatomical and electronic microscope and stored in 80% alcohol for subsequent use.

### 4.2. RNA Isolation and Molecular Cloning

Total RNA from all samples of healthy *P. monodon* was obtained using the TRIzol reagent (Invitrogen, Carlsbad, CA, USA) and the HiPure Fibrous RNA Plus Kit (Megan, Guangzhou, China). The purity and quantity of total RNA were calculated by measuring the ultraviolet absorbance ratio at 260/280 nm using a NanoDrop 2000 device (Thermo Scientific, Waltham, MA, USA) and 1% agarose gels. The cDNA synthesis was performed using the Evo M-MLV RT Kit with gDNA Clean for qPCR (AG, Hunan, China), and the product was stored at −20 ℃ until subsequent use. Primers were designed using Primer Premier 5.0 software based on partial sequences retrieved from the *P. monodon* transcriptome library analyzed in our laboratory (Table 1). The *PmDsx* open reading frame (ORF) was expanded by PCR technology. The final PCR products were cloned into the pMD-19T vector (Takara, Kyoto, Japan), sequencing a positive monoclonal colony.

### 4.3. Bioinformatics Analysis of PmDsx

The sequencing results were identified using the National Center for Biotechnology Information (NCBI) BLAST search program (http://www.ncbi.nlm.nih.gov/blast, accessed on 12 October 2021). The ORF and its corresponding amino acid sequences of PmDsx were predicted using the ORF Finder (https://www.ncbi.nlm.nih.gov/orffinder, accessed on 12 October 2021). ExPASy-PROSITE (https://prosite.expasy.org/, accessed on 12 October 2021) was used to analyze the functional domain of protein prediction. N-glycosylation sites and phosphorylation sites were predicted by NetNglyc 1.0 server and NetPhos3.1 server Software (https://services.healthtech.dtu.dk, accessed on 13 October 2021). Multiple sequence alignments and phylogenetic analysis of the predicted PmDsx protein with known Dsx proteins were analyzed using the HiPlot online software (www.hiplot.com.cn/, accessed on 13 October 2021). The three-dimensional protein structure of PmDsx was constructed by SWISS-MODEL online software.

### 4.4. Real-Time Quantitative PCR Analysis

The qRT-PCR amplification was performed on a Roche 480 machine using SYBR^®^ Premix Ex Taq™ II (Takara, Kyoto, Japan) as the fluorescent dye. The translation elongation factor 1-alpha (EF1α) was used as a reference gene, and all reactions were performed in triplicate. The PCR assay was carried out in a total volume of 12.5 μL, containing 6.25 μL of 2 × SYBR Premix Ex Taq (Takara, Kyoto, Japan), 1 μL of the diluted cDNA, 0.5 μL of each primer (10 μM), and 4.25 μL of sterile distilled H_2_O. The cycle conditions were described as follows: 95 °C for 3 min, followed by 40 cycles of 95 °C for 5 s, 60 °C for 30 s, and 72 °C for 30 s. Amplification curve analysis of the amplification products at the end of each thermal cycling reaction was performed to confirm the specificity of the amplification and to ensure successful amplification and detection, and fold change for the target gene relative expression of mRNA relative to controls was calculated using the 2^−^^△△Ct^ method [45].

### 4.5. RNA Interference

The double-stranded RNA (dsRNA) for PmDsx and pEGFP-N1 were used as a linearized DNA template for in vitro transcription reactions to synthesize dsRNA by the T7 RiboMAX™ Expressed RNAi System Kit (Promega, WI, USA). The specific primers containing the T7 promoter were designed for transcriptional amplification, and dsRNA formation was quantified by a 1.2% agarose gel electrophoresis. The final dsRNA product was stored at −80 °C. The synthetic dsRNA was diluted to 3 μg/μL with PBS. Before the formal injection experiments, we performed a 48-h pre-experiment to test the silencing efficiency. Each shrimp was injected intramuscularly into the second abdominal segment. Based on the results, we determined that 3 μg of dsRNA was injected per gram of shrimp. The experiment consisted of three groups (30 shrimps per group): the dsPmDsx injection group (experimental group), the PBS injection group (negative control group), and the dsGFP injection group (positive control group). The shrimp were injected every two days, and the testis of three shrimps from each group was collected at 0, 2, 4, and 6 days before each injection. Then, the total RNA was extracted from the testis of shrimps by the same method mentioned above.

### 4.6. Promoter Cloning and Plasmid Construction

The Transcription Factor Binding Site Prediction (TFBS)-JASPAR database (http://JASPAR.genereg.net/, accessed on 23 November 2021) and ALGGEN-PROMO (http://alggen.lsi.upc.es, accessed on 23 November 2021) were used to search for potential binding sites to determine the potential function of the *PmDsx* binding site on the core *PmIAG* promoter. The full-length ORF of *PmDsx* was amplified and inserted into the pCMV-C-myc vector (Beyotime, China) containing the Hind III/Bgl I site, called pCMV-PmDsx. In addition, four truncated mutants from the *PmIAG* promoter were designed based on the location of the *PmDsx* predicted binding site. These fragments have a common 3’-end at −3 bp (translation start codon defined as + 1) and 5’-ends at −1629, −1193, −938, and −371 were amplified using specific primers (Table 1). The vectors were named pGL3-basic-Luc1, pGL3-basic-Luc2, pGL3-basic-Luc3, and pGL3-basic-Luc4, and each fragment was cloned into the pGL3-Basic firefly luciferase expression vector (Promega, USA) for the dual-luciferase reporter assay. The pRL-TK luciferase reporter vector was considered an internal control.

### 4.7. Dual-Luciferase Reporter Assay

The luciferase activity was measured according to the protocol for the Dual-Luciferase Reporter Assay System (Promega, Madison, WI, USA). The pCMV-PmDsx overexpression vector was separately cotransfected into HEK293 cells with a series of constructs containing luciferase reporter genes driven by different species of *P. monodon IAG* promoters. Cells were collected and lysed in reporter lysis buffer (Promega, USA) after 48 h of transfection. Then, 100 μL of LARII firefly luciferase assay reagent was added to 20 μL of the cell lysate to be measured. The mixture was then quickly placed into a luminescence detector (Thermo Fisher Scientific, Waltham, MA, USA) and the fluorescence value emitted by the substrate excited by firefly luciferase was read A. An amount of 100 μL of 1× Stop & Glo renilla luciferase assay was then added, and the fluorescence value emitted by mesothelial luciferase of the test samples was read B. The ratio of each sample Ratio = A / B value is the relative luciferase activity of cells transfected with the expression plasmid constructed for the fragment. If the ratio of the experimental group is significantly higher than the ratio of the empty group, it proves that the target gene is activated. All experiments were repeated three times independently.

### 4.8. Statistical Analysis

The statistical significance of data differences between genders was assessed using one-way ANOVA with GraphPad Prism 8 software. Gene expression differences between different tissues were evaluated using GraphPad Prism 8 software. Data about the luciferase activities were analyzed as mean + SE, and *p* < 0.05 was statistically significant.

## 5. Conclusions

In summary, this study characterized the *Dsx* gene in *P. monodon*, which exhibits a male-biased expression pattern in different tissues. The putative *Dsx* binding site was identified in the promoter region of the male sexual differentiation effector gene *PmIAG*. In addition, *PmDsx* positively regulated the transcription of the *PmIAG* gene by binding directly to a specific binding site in the *PmIAG* gene promoter, suggesting that *PmDsx* may be an upstream regulator in the process of sexual differentiation in shrimp. Our results provide a reference for exploring the molecular mechanisms of sex determination and differentiation in *P. monodon* and even crustaceans.

## Figures and Tables

**Figure 1 ijms-23-12701-f001:**
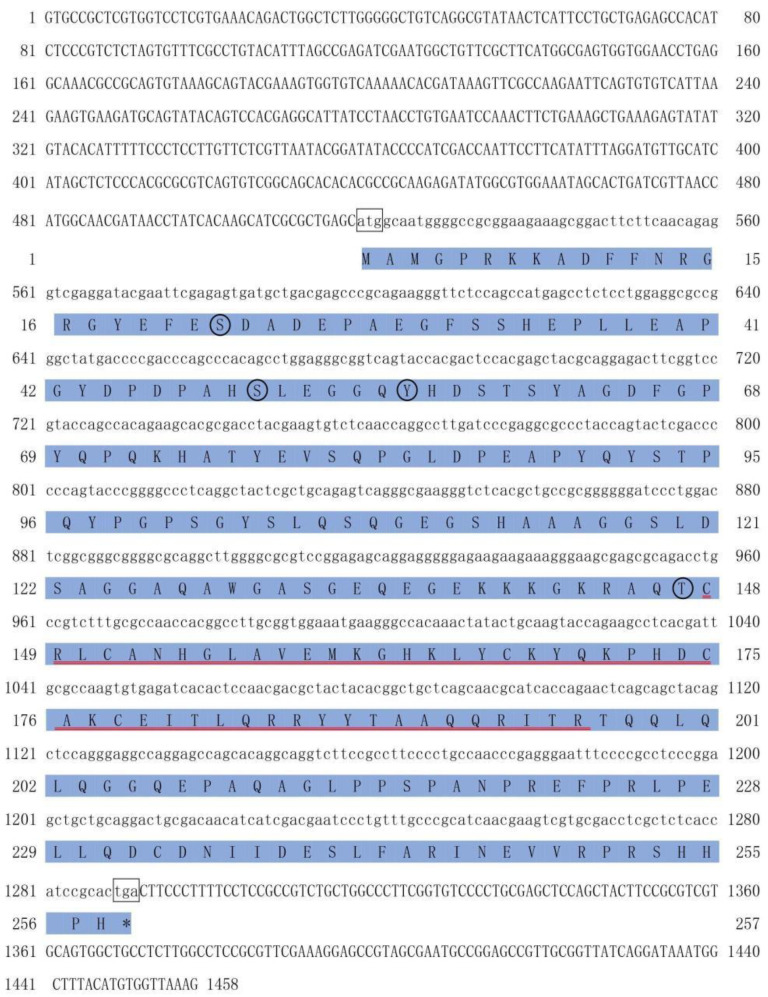
Nucleotide and amino acid sequences of *PmDsx*. The initiation codon (ATG) and a termination codon (TAA) are boxed in black, the (ORF) is shaded in blue, the DM domain is underlined by the red line, and the phosphorylation sites are circled in black.

**Figure 2 ijms-23-12701-f002:**
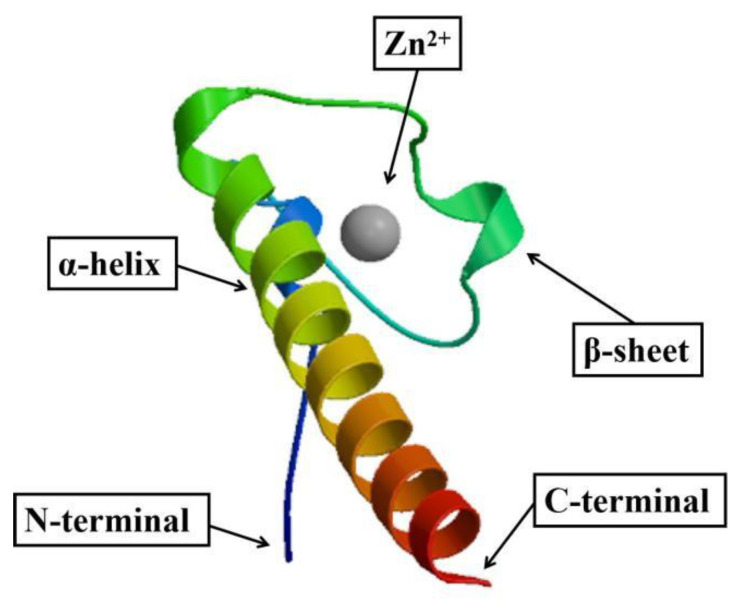
Three-dimensional protein structure of PmDsx. The protein of PmDsx is composed of zinc finger structures, containing α-helix, anti-parallel β-sheets, and Zn^2+^.

**Figure 3 ijms-23-12701-f003:**
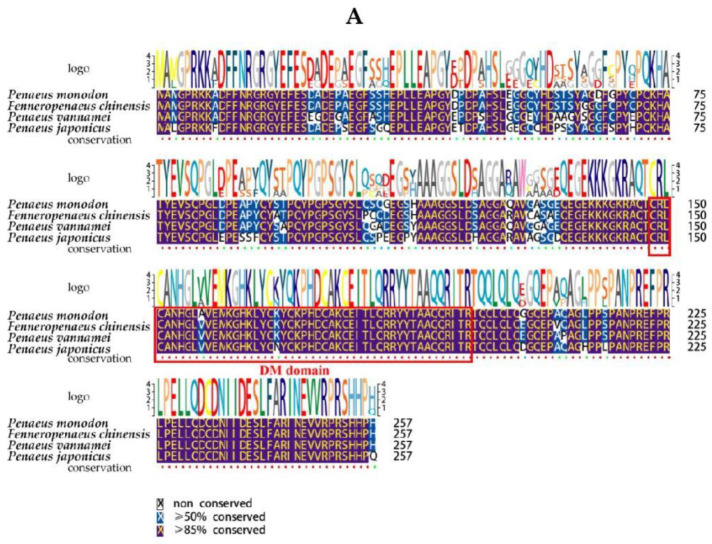
Multiple sequence alignment and phylogenetic analysis of the proteins of PmDsx and other species. (**A**)The Dark purple areas are exactly the same, and the blue areas indicate that the sequence has more than 50% similarity. The DM domain is boxed in black. (**B**) The neighbor-joining phylogenetic tree of PmDsx on their amino acid sequences in different organisms was constructed by MEGA 6.0, and the confidence in each node was assessed by 2000 bootstrap replicates. The position of the *P. monodon* Dsx protein on the phylogenetic tree is marked by a red triangle. Species clustered into one group are indicated by boxes of the same color.

**Figure 4 ijms-23-12701-f004:**
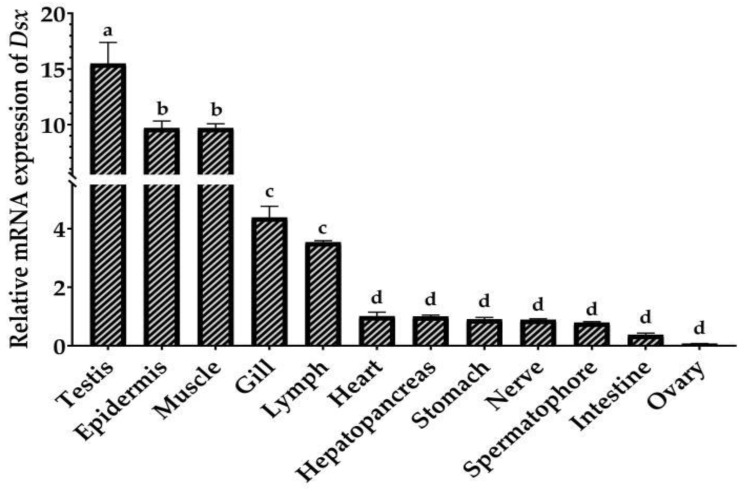
Relative expression of *PmDsx* in different tissues of *P. monodon*. Relative expression levels of *PmDsx* in 12 tissues were detected by qRT-PCR, the statistics were calculated by the 2^−^^ΔΔCt^ method, and expressed as mean ± standard deviation (*n* ≥ 3). The same letters mean the difference is not significant, and different letters mean the difference is significant (*p* ≤ 0.001).

**Figure 5 ijms-23-12701-f005:**
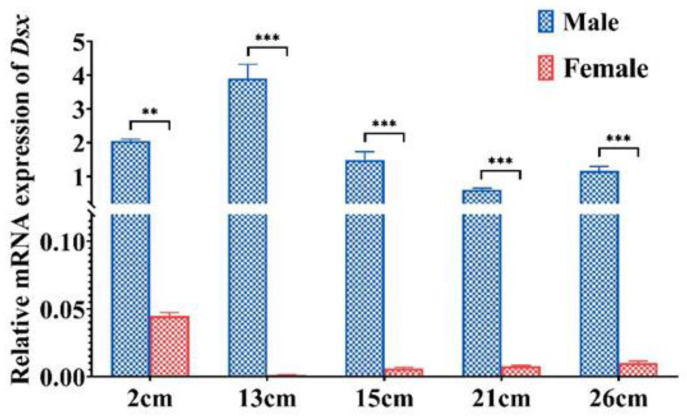
Expression of *PmDsx* in males and females at different developmental stages. qRT-PCR was used to detect differential expression in different developmental stages (2 cm: post larval; 13~15 cm: gonadogenesis stage; 21~26 cm: gonadal maturation stage). Data were calculated by the 2^-ΔΔCt^ method and expressed as mean ± standard deviation (*n* ≥ 3), and different asterisks indicate statistically significant differences (*** is *p* ≤ 0.0001, ** is *p* ≤ 0.001).

**Figure 6 ijms-23-12701-f006:**
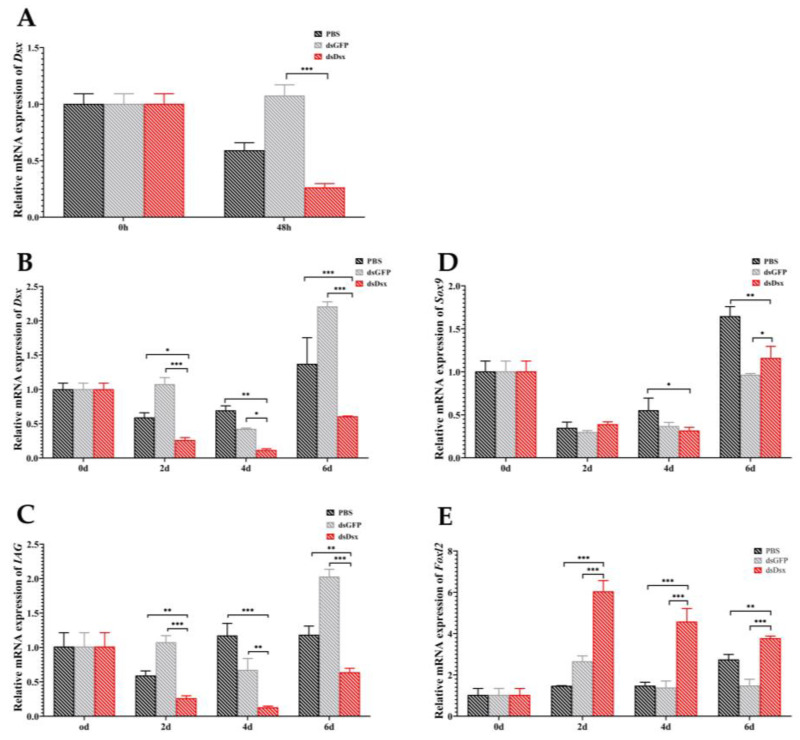
Time-dependent silencing effect of *PmDsx*, *PmIAG*, *PmSox9*, and *PmFoxl2* in testis. The interference efficiency of *PmDsx* (**A**) and the expression of *PmDsx* (**B**), *PmIAG* (**C**), *PmSox9* (**D**), and *PmFoxl2* (**E**) at 0, 2, 4, and 6 days after dsPmDsx injection were analyzed by qRT-PCR. The PBS and dsGFP groups were the control group. Three individuals were pooled as one sample and three replicates were used for analysis. The bars represent the mean ± standard deviation (*n* = 3), and different asterisks indicate statistically significant differences (*** is *p* ≤ 0.0001, ** is *p* ≤ 0.001, * is ≤ 0.01).

**Figure 7 ijms-23-12701-f007:**
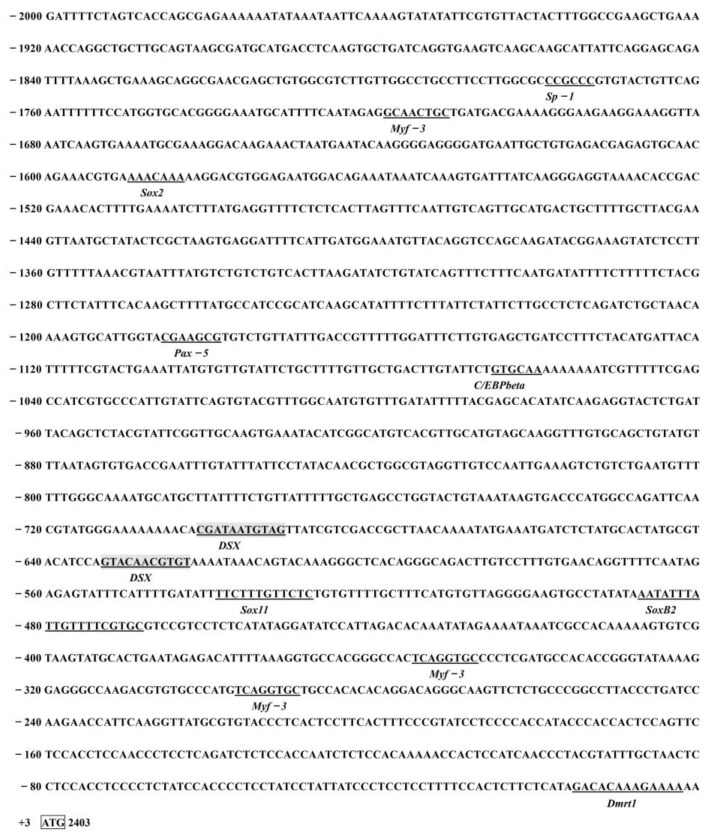
Sequence analysis and fragment amplification of *IAG* promoter in *P. monodon*. Bioinformatic prediction of potential cis-regulatory elements of transcription factors within the *IAG* gene promoter of *P. monodon*. Binding sites for binding various transcription factors are underlined, translation initiation sites are circled by boxes, and *Dsx* binding sites are shaded in gray.

**Figure 8 ijms-23-12701-f008:**
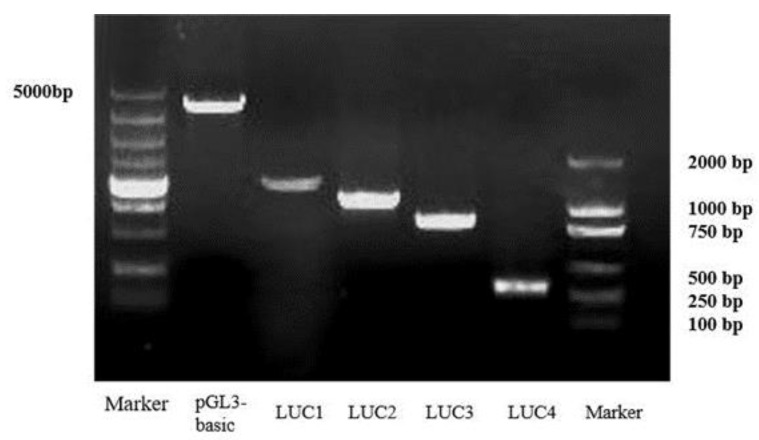
Amplification of different segments of the *P. monodon IAG* promoter sequence. The PCR amplification products were verified and purified by agarose gel electrophoresis.

**Figure 9 ijms-23-12701-f009:**
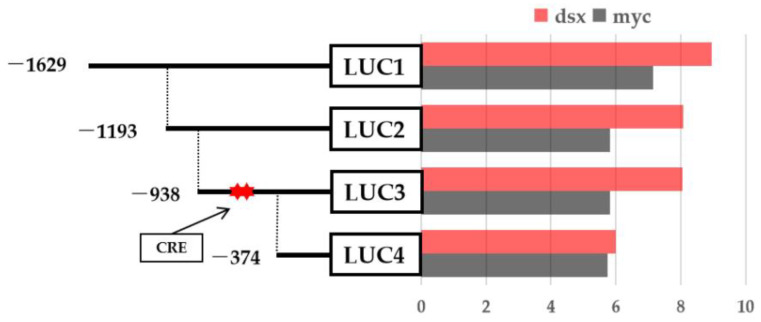
Effects of *PmDsx* overexpression on the activity of different 5’ flanking regions of the *IAG* promoter in *P. monodon*. The red pentagram represents the binding site of *PmDsx* in the *PmIAG* promoter region. The data represent the means ± SE (*n* = 3). The different symbols above the error bars in the figure indicate statistical differences at *p* < 0.05, as determined by one-way ANOVA followed by the post hoc test compared with the control.

**Table 1 ijms-23-12701-t001:** Primers and sequences used in the study.

Primer	Primer Sequence(5′–3′)	Purpose
PmDsx-F	GTGGAACCTGAGGCAAACG	ORF Validation
PmDsx-R	AGACGGCGGAGGAAAAGG	ORF Validation
PmDsx-qF	TCCAACGACGCTACTACACGG	qPCR
PmDsx-qR	TTGATGCGGGCAAACAGG	qPCR
dsPmDsx-T7-F	TAATACGACTCACTATAGGGAGAGTACCAGCCACAGAAGCACG	RNAi
dsPmDsx-T7-R	TAATACGACTCACTATAGGGAGAAGACGGCGGAGGAAAAGG	RNAi
dsPmDsx-F	GTACCAGCCACAGAAGCACG	RNAi
dsPmDsx-R	AGACGGCGGAGGAAAAGG	RNAi
dsGFP-T7-F	TAATACGACTCACTATAGGGAGACCGACAAGCAGAAGAACGGCATCA	RNAi
dsGFP-T7-R	TAATACGACTCACTATAGGGAGATCACGAACTCCAGCAGGACCATGTGA	RNAi
dsGFP-F	CCGACAAGCAGAAGAACGGCATCA	RNAi
dsGFP-R	TCACGAACTCCAGCAGGACCATGTGA	RNAi
PmSox9-qF	GCGAGGGACTTGGTAAATGTG	qPCR
PmSox9-qR	TGGTGGCTAGGATTGGTCTGA	qPCR
PmIAG-qF	GCCTTGAATCCGATGCGATAT	qPCR
PmIAG-qR	GCAGCACTCATCCTGTACGTTGT	qPCR
PmFoxl2-qF	AGGGAAGGGGAATTTCTGG	qPCR
PmFoxl2-qR	AAGCGTCGGGGTAGGTGTA	qPCR
EF-1α-qF	AAGCCAGGTATGGTTGTCAACTTT	Reference gene
EF-1α-qR	CGTGGTGCATCTCCACAGACT	Reference gene
LUC1-F	TAAAGGTGCCACGGGCCACTCAGG	Sequence Validation
LUC1-R	CATTTTTTTCTTTGTGTCTATGAG	Sequence Validation
LUC2-F	CAAGTGAAATACATCGGCATGTCAC	Sequence Validation
LUC2-R	CATTTTTTTCTTTGTGTCTATGAG	Sequence Validation
LUC3-F	ATTGGTACGAAGCGTGTCTGTTAT	Sequence Validation
LUC3-R	CATTTTTTTCTTTGTGTCTATGAG	Sequence Validation
LUC4-F	GATGAATTGCTGTGAGACGAGAGT	Sequence Validation
LUC4-R	CATTTTTTTCTTTGTGTCTATGAG	Sequence Validation

## Data Availability

Data are contained within the article or Appendix A.

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
