# Peer review of "Identification and Expression Analysis of Dsx and Its Positive Transcriptional Regulation of IAG in Black Tiger Shrimp (Penaeus monodon)"

_ijms, 2022, doi:10.3390/ijms232012701_

Round 1

Reviewer 1 Report

A great deal of attention has recently been paid to the molecular genetic pathways that lead to sex differentiation in aquaculture species and their potential manipulation through RNA interference. In this manuscript, Wei et al., have identified and cloned the Doublesex (Dsx) gene from Penaeus monodon, an important aquaculture species in Asia. The cloned PmDsx cDNA encoded for a 257 aa protein had high similarity to other shrimp Dsx orthologs. PmDsx mRNA expression was analysed in various tissues and different developmental stages.  dsRNA mediated RNAi of PmDsx in-vivo decreased the expression of IAG, but increased PmFoxl2. The key Dsx binding site of the PmIAG gene promoter were mapped using truncated regulatory domain regions pGL4 Luciferase Reporter constructs.

Despite the well-studied role of the Dsx gene in sex differentiation among arthropods such as Drosophila, Daphnia, etc, only a few orthologs of the Dsx gene have been identified from decapod species of economic importance (Macrobrachium, Scylla). As the first report of Dsx gene identification and cloning from Penaeus monodon, this study is significant considering the important role of the species in aquaculture in this part of the world.

The manuscript is well written, the methods used are typical for this sort of research, and the results are clearly presented. It is, however, requested that the authors provide some additional points that can help improve the manuscript.

Comments:

1. Figure 3 Multiple Sequence alignment:  Amino acid sequence alignment of the PmDSx and other orthologs lack details. From the current figure, It is very difficult to obtain detailed amino acid sequence information and residue conservation details. The authors are requested to provide a conventional multiple sequence alignment in the standard format with the amino acid sequences indicated by their one letter abbreviation. In the alignment, the authors are requested to clearly mark the conserved domains, motifs and key amino acid residues etc.

2. Figure 3 Phylogenetic tree:  The phylogenetic tree can be mode more intuitive by including DM-domain containing protein sequences like DMRT93B, DMRT11ER, DMRT4, DMRT2, DSX1, DSX2 from other invertebrates. Such a tree will illustrate the evolutionary relationship much better. Also include in the figure legend details of the phylogenetic method, number of bootstrap replicates and software used. The authors are also requested to include the bootstrap values at the nodes of the phylogenetic tree.

3. Figure 4 : Please provide a standard bar chart indicating the relative expression of PmDx in different tissues. The current web diagram does not seem to accurately indicate relative gene expression analysis with appropriate statistical parameters for the data derived from RT-qPCR studies.  Also provide appropriate statistical details like error bars, details of analysis used for significance, statistical significance details etc.

4. Figure 5: The authors do not mention what tissues were used in the analysis of expression of PmDsx mRNA by qPCR at different developmental stages of male/female of P. monodon.

5. Figure 6: In Figure 6A, the PBS control injected shrimp also shows drastic downregulation of mRNA expression of Dsx at 48 hr. But the dsGFP injected shrimp does not show downregulation. In the analysis of statistical significance, was the change in expression levels of PBS/Dsx injected as well as PBS/dsGFP analysed. It would be great if the authors can check on these data and comment on the downregulation of Dsx expression in PBS control injected sample.

6. Section 4.5 RNA interference – Scrambled siRNA control missing in the experiments: It is well known that siRNA is capable of interacting with a variety of other pathways non-specifically and initiating off-target effects. So, an siRNA scrambled control was essential in the experiments. The authors are expected to comment on this.

Reviewer 2 Report

The abstract from Wei et al., delves into the regulatory role of the Dsx gene in male Black Tiger Shrimp. This is a well-paced work clearly showing how this gene is differentially expressed in male and female individuals, and how disruption of Dsx leads to the differential expression of several sex-related genes, and culminates by proving the direct regulation of Dsx over the insulin-like androgenic gland hormone gene, which has been postulated to be critical for male differentiation. There are, however, a few issues I would like to comment on in order to improve the quality of this work.

L18-19: The authors state that “transcriptional regulation of the Dsx gene is largely unknown in decapods”. I am not sure if they refer to transcriptional regulation by Dsx, or regulation of the Dsx gene as they state, as they do not approach this in their work.

L51: Please provide the full name for the STAT acronym.

L93: I was missing some explanation about the relevance of phosphorylation sites in Dsx in this section.

The quality of the figures can be improved; this is especially important for Fig. 2 and 6, which cannot withstand a proper zoom.

Fig. 2: I could not find which method and software were used for generating the structural model for Dsx, please provide this information.

Fig. 3: Change Penaeus chinensis in figure 3 to Fenneropenaeus chinensis to avoid confusion.

Fig. 5 and 6: Please provide a full explanation of the different p values when using different asterisks.

Fig. 6: It seems that PBS injection has a significant impact on Dsx gene expression as well as on Sox9 and IAG. I am curious about what the authors can comment about this.

L148: Even though this information is included elsewhere in the MS, please elaborate a bit on the interference method used to facilitate the reading and to improve the flow.

L153: It is hard to conclude looking at the data that Sox9’s expression is decreased because of the interference, since there are no significant differences at the 2 days stage, and also no differences with the GFP control after 4 days, so this decrease seems to be highly affected by the method. The effects are at least not as evident and consistent as in their previous work with PmDMRT11E, so I would suggest the authors to treat this result with caution, and maybe refer to this as a potential downregulation which needs further study, rather than a proven decrease in expression caused by Dsx interference.

L170: Reference to Fig.7 is a bit confusing; it seems to reference the truncated mutants instead of the location of the binding sites.

Fig. 8: The results of the PCR are not really adding much. I don’t consider this figure necessary.

L183-184: Might be an erratum but referring to “each of the PmDsx constructs” sounds like if there were more constructs than the wild type PmDsx.

Fig. 9: There seems to be a mistake with the naming of the different promoter constructs, since in Fig. 8 the four constructs are named LUC 1 to 4, and in Fig. 9, they are named LUC 2 to 5, with the 938 construct apparently receiving two different names in both figures, please double-check.

L230, L234, L384: Although its use is quite extended, the term “sex” is often preferred to “gender” in non-human animal models. This is especially the case when using expressions such as “gender specific role” which have different meanings in other scientific fields.

Table 1: The primers used to generate the different promoters tested for the luciferase assays seem to be missing.

I think this work would reach a more rounded up conclusion if the authors would acquire more conclusive data about the two putative binding sites in the PmIAG promoter by performing an additional experiment. Either an EMSA showing binding to both putative binding sites, or two additional truncations of the promoter that exclude just one of the binding sites, and the region that has both (for example truncating it around -640 and around -560), would clearly delimitate the regulatory sites and show if both binding sites are functional.
